# LOQTEQ^®^ VA Periprosthetic Plate—A New Concept for Bicortical Screw Fixation in Periprosthetic Fractures: A Technical Note

**DOI:** 10.3390/jcm11051184

**Published:** 2022-02-23

**Authors:** Clemens Kösters, Daniel den Toom, Sven Märdian, Steffen Roßlenbroich, Sebastian Metzlaff, Kiriakos Daniilidis, Jens Everding

**Affiliations:** 1Klinik für Orthopädie, Unfall-und Handchirurgie, Maria-Josef-Hospital Greven, 48268 Greven, Germany; daniel.toom@mjh-greven.de; 2Centrum für Muskuloskelettale Chirurgie, Charité Berlin, 13353 Berlin, Germany; sven.maerdian@charite.de; 3Klinik für Unfall-, Hand-und Wiederherstellungschirurgie, Universitätsklinikum Münster, 48149 Münster, Germany; steffen.rosslenbroich@ukmuenster.de (S.R.); jens_everding@hotmail.com (J.E.); 4Klinik für Orthopädie und Unfallchirurgie, St. Joseph Krankenhaus Berlin, 12101 Berlin, Germany; sebastian.metzlaff@sjk.de; 5Orthopädie Traumatologie Centrum Regensburg, 93049 Regensburg, Germany; daniilidis@otc-regensburg.de

**Keywords:** plate osteosynthesis, periprosthetic fracture, hinge plate, bicortical screw fixation

## Abstract

Internal fixation using angle stable plates is the treatment standard in periprosthetic fractures around stable implants. To provide instant postoperative full weight-bearing, bicortical screw fixation is advisable but often surgically demanding. This work presents the first clinical results of the LOQTEQ^®^ VA Periprosthetic Plate (aap Implantate AG, Berlin, Germany), a new plate system to simplify screw placement around implants. This plate system uses insertable hinges that allow for variable angle screw anchorage. Data of 26 patients with a mean age of 80 years and a mean follow-up of 13.9 months were retrospectively collected. Patients were clinically and radiologically examined. Bony union was achieved in 14 out of 15 patients with no signs of non-union or implant loosening. One patient, however, presented with implant failure. Clinical scores demonstrated acceptable results. Since the hinge plates are easy to apply, the first results are promising.

## 1. Introduction

Periprosthetic fractures following total hip (THA) or knee (TKA) arthroplasty represent a serious complication and are the third most common cause of revision surgery [1,2,3]. Their treatment demands special attention to patient management and requires profound expertise in both trauma surgery and revision arthroplasty techniques [1,4,5,6]. In the elderly, stabilization of such fractures may be difficult in the presence of osteoporosis, previous surgeries, and a significant risk profile due to severe comorbidities [4]. Due to the demographic change in industrial countries with an aging but still-active population combined with the increasing life expectancy and prolonged survivorship of the prostheses markedly increases the rate of periprosthetic fractures. Several classification systems have been introduced to guide surgeons to optimized treatments [7,8]. Duncan and Haddad proposed the UCS Classification, which is applicable regardless of the bone that is broken or the joint involved [9]. 

The most crucial aspect in all established treatment algorithms is determining whether the prosthesis is stable or not [7,8]. To date, open reduction and internal fixation (ORIF) using modern locking plate systems is regarded gold standard for periprosthetic fractures around stable implants. Numerous different plates and additional devices have been introduced during the last years that allow fixation adjacent to a prosthetic stem and provide sufficient stability for successful bone healing [10,11]. Bicortical screw fixation provides the highest stability compared to mono-cortical screws or cable cerclages [12,13]. However, there is also biomechanical evidence that the type of bicortical fixation (i.e., dimension of the screws used) has a significant impact on failure modes [12]. Especially in cases of periprosthetic fractures around bulky implants, bicortical screw placement can be demanding or even impossible. On this account, special implants and techniques have been designed to facilitate periprosthetic implant anchorage. The locking attachment plate (LAP) (Depuy-Synthes, Solothurn, Switzerland) represents the first device that was exclusively developed to allow periprosthetic locking screw anchorage (Figure 1). 

This plate can be attached to any 4.5 mm locking compression plate (LCP) (Depuy Synthes, Solothurn, Switzerland) and offers four additional 3.5 mm locking screws around the femoral stem. However, inherent to the system, the screws can be placed monoaxial only. Thus, the screw might fail to bypass the prosthesis or might even damage it. Currently, a new system has been introduced into the market that might solve these issues. This plate system uses attachable hinges that allow for variable angle screw anchorage (LOQTEQ^®^ VA Periprosthetic Plate, aap Implantate AG, Berlin, Germany) (Figure 2).

This paper aims to describe the surgical technique and first clinical results of a new plate system which was designed to simplify bicortical screw placement around implants located in the femur.

## 2. Patients and Methods

Twenty-six patients with periprosthetic femoral fractures using the LOQTEQ^®^ VA Periprosthetic Plate were included in this retrospective study. Preoperative CT scans allowed the exact analysis of the fracture, bone quality, and stability of the prosthesis in order to plan the surgical treatment. Furthermore, taking the results of the CT-scan into consideration, classification of the fracture according to the Unified Classification System (UCS) was performed. All patients were operated on by two senior surgeons (C.K. and J.E.). Table 1 shows the study characteristics of our patient collective.

### 2.1. Surgical Technique and Implant 

Positioning of the patient and surgical approach is similar to the surgical technique of a distal femoral plate. A standard incision via a lateral approach was performed in all cases. Minimal invasive implantation using a carbon aiming device is optional. After fracture reduction, temporary fixation of the plate with K-wires, and fluoroscopic control of the ideal plate position, the first variable angle screws are fixed in the distal part of the plate. The new plate system provides two options in the distal metaphyseal part and 8 options in the proximal part of the plate to place special hinges which each has two holes for variable angle 3.5 mm locking screws that can be directed at an angle of 15° in all directions. (Figure 3).

The hinges are placed with a special device to allow a minimally invasive application. Once the hinge is clicked on the lateral part of the plate, it is possible to rotate it at an angle of 45° around the plate. In the desired position, it is then fixed with a central screw that compresses the hinge around the plate. The possibility to rotate the hinge around the plate allows even more possible directions for each variable angle screw. Once the hinge is positioned, the surgeon has two options to drill a 2.7 mm hole – either in using a 15° cone or with freehand drilling (Figure 4).

Screws are inserted and fixed with a limited torque screwdriver (3.5 Nm). Optionally the central hole of the plate could be filled with a mono-cortical blunt screw in the area of the placed hinges. Furthermore, the plate provides the option to place a “click-grommet” for the use of cable wire cerclages of all different sizes available on the market. 

Figure 5 shows an example of an internal fixation using the LOQTEQ^®^ VA Periprosthetic Plate. 

### 2.2. Intra- and Perioperative Data

Intra- and perioperative data were extracted from the medical records, including length of operation, implantation of an additional anterior plate, duration of hospital stay, and complications. Double plating (additional anterior or medial plate) was performed in the case of intraoperatively detected osteoporotic bone based on the surgeon’s impression and comminuted fractures. Table 2 demonstrates the intra- and postoperative data.

### 2.3. Clinical Scores and Radiological Examinations

Full weight-bearing was allowed for every patient postoperatively [14]. All patients treated with the new periprosthetic device were examined clinically and radiologically. Data for pain (VAS) and functional scores (WOMAC-Score, Harris-Hip-Score, or Knee-Society-Score) were collected. The radiographic examination included a.p. and lateral views of the femur and adjacent joints (hip/knee) to assess bone union, implant positioning (incl. screw positioning) and complications like secondary loss of reduction (fragment angulation and fragment torsion) or implant failure. Evaluation of the radiographs was performed by two senior physicians. Any discrepancies concerning the radiological assessment were resolved by a third independent senior radiologist. 

### 2.4. Statistics

Statistical analysis was conducted using IBM SPSS^®^ Statistics 26 (IBM Corporation, Somers, NY, USA) and Microsoft Excel 16.51. 

## 3. Results

### 3.1. Epidemiological Data

Twenty-six patients were included in this study. All patients underwent surgery for the initial prosthesis due to osteoarthritis. A total of 15 out of those 26 patients were available for the clinical follow-up examination; 1 patient presented with implant failure to subsequently undergo revision surgery but was lost to follow-up afterward. Eleven patients lacked follow-up due to incorrect contact details (1 patient), frailty (4 patients), or death because of age and multimorbidity (6 patients). All six patients did not undergo revision surgery. The mean clinical follow-up time was 13.9 months (SD 12). The mean age of the 26 patients was 80 years (SD 11). According to the UCS Classification, all 26 periprosthetic fractures were classified as follows: 8x V.3-B1, 6x V.3-D, 5x IV.3-B1, 3x IV.3-C, 3x V.3-C, and 1x V.3-B3. After the radiological assessment of the implant stability, an additional intraoperative stability check was performed. The average time interval between primary prosthesis implantation and occurrence of the fracture was 16 years (SD 11), whereas the mean time interval between injury and operative treatment was 2 days (SD 2). In all cases, the mechanism of injury was a low-energy trauma.

### 3.2. Intraoperative Technical Data

The mean operation time was 165 minutes. In all cases, intraoperative radiological examinations showed acceptable implant positioning. Limb length, bone alignment, and rotation were adequately reconstructed in all patients. Double plating (additional anterior or medial plate) was performed in 8 patients.

### 3.3. Postoperative Clinical and Radiological Follow-Up

The examination revealed a mean Harris Hip Score of 63 points (SD 27), a mean Knee Society Score of 118 points (SD 60) and a mean WOMAC Index of 38 points (SD 23). At the time of follow-up with a mean VAS of 2.6, three patients had no residual pain (VAS 0). Eight patients reported occasional mild pain (VAS 2–4) and four patients described moderate pain (VAS 5–6). Radiologic bony union was achieved in 14 patients at the follow-up examination.

### 3.4. Complications

One patient required a surgical revision due to a fall on the ward, with subsequent secondary dislocation of the fracture seven days after the index operation. In the following revision surgery, the lateral plate was combined with an additional anterior 3.5 mm plate to increase the stiffness of the osteosynthesis. Two patients required reoperation due to hematoma.

Another patient that was treated with an additional anterior plate in the initial surgery suffered from implant failure (plate fracture) without a trauma 6 months after osteosynthesis. 

## 4. Discussion

This study presents a newly developed plate system for the stabilization of periprosthetic femur fractures. Hinges are attached to the plate without blocking a locking hole, offering two main advantages. First, the angle of each hinge plate is freely adjustable before fixing the hinge to fit the anatomy of the patient. Second, two 3.5 mm variable angle screws can be inserted in each of the hinges. With these options, the new implant system allows an easier bicortical locking around hip stems, particularly with demanding geometries.

A biomechanical study demonstrated that the LOQTEQ^®^ VA periprosthetic plate has superior characteristics regarding axial stiffness and the number of cycles to failure compared to the LAP. The plate showed a 43% higher axial stiffness and resisted 20% more load-cycles until construct failure [15]. This becomes particularly important in the elderly patient, who suffers from osteoporosis and impaired fracture healing. In a clinical study conducted by Dumpies et al. (2012), insufficient radiological fracture healing occurred in 33.3% after LAP implantation [16]. Although our study included only 15 examined patients, 14 of them showed bony union except for one case of implant failure. Compared to the LAP, the LOQTEQ^®^ VA Periprosthetic Plate, especially combined with an additional plate, provides higher stiffness, thus making postoperative full weight-bearing possible [17,18]. This might have contributed to our excellent consolidation rates. 

In a systematic literature review, Stoffel et al. (2016) evaluated the influence of operation technique and implant used in the treatment of periprosthetic and interprosthetic fracture fixation. Thirty-one studies with a total of 457 patients were evaluated. In all cases, osteosynthesis was performed with a locking plate. The results showed a fixation failure of 8.1%, reoperation rate of 10.1%, a non-union rate of 1.1%, and an infection rate of 2.6% [19]. These rates are comparable to our study but will have to be confirmed by larger clinical trials in the future. 

The functional scores of the present study revealed a Harris Hip Score of 63 points (SD 27), Knee Society Score of 118 points (SD 60), and a mean WOMAC Index of 38 points (SD 23). Moreta et al. referred to a mean Harris Hip Score of 59–73 in comparable literature [20]. Reviewing the outcome after periprosthetic fractures, Aprato et al. concluded a mean WOMAC Index of 39 points [21]. Our results appear to be similar to the literature; there is, however, a high range in all of our functional scores, which might be due to the high range of our patients’ age. Furthermore, the poor results of functional scores in periprosthetic fractures need to be assessed in relation to the severity of the fracture, general frailty, and multimorbidity of our geriatric patient cohort. A mean VAS of 2.6 ± 1.9 complements our good clinical results and equals comparable studies as well [22].

This study has some inherent limitations. First of all, there is a lack of a comparison group. Furthermore, the retrospective design of this study is generally prone to information bias of the collected data. However, since the information is mostly noted by default, e.g., operation time or the exact surgical procedure, the bias is of lesser consequence. Since it is a retrospective case series, no significant conclusion can be drawn directly from this study.

The rather short mean follow-up of 13.9 months further complements this issue, so more studies, especially in randomized controlled design, will have to be carried out. Nevertheless, considering the age and multimorbidity of the study population, a long-term follow-up will prove impossible, as reflected in our high dropout rate.

## 5. Conclusions

According to our experience regarding implant fixation in periprosthetic fractures, using bicortical screws around well-fixed implants proves there is more comfort in utilizing the new variable angle hinge plate. The first clinical results are promising. To date, only one case of implant failure was observed, however, a higher number of patients to gain more experience with the system as well as randomized controlled trials are needed to prove the superiority of the new periprosthetic hinge plate over the standard implants.

To conclude, the LOQTEQ^®^ VA Periprosthetic Plate shows encouraging first results. The new implant system seems to be beneficial for the treatment of periprosthetic fractures especially due to the increased flexibility in bicortical screw placement and the increased stability.

## Figures and Tables

**Figure 1 jcm-11-01184-f001:**
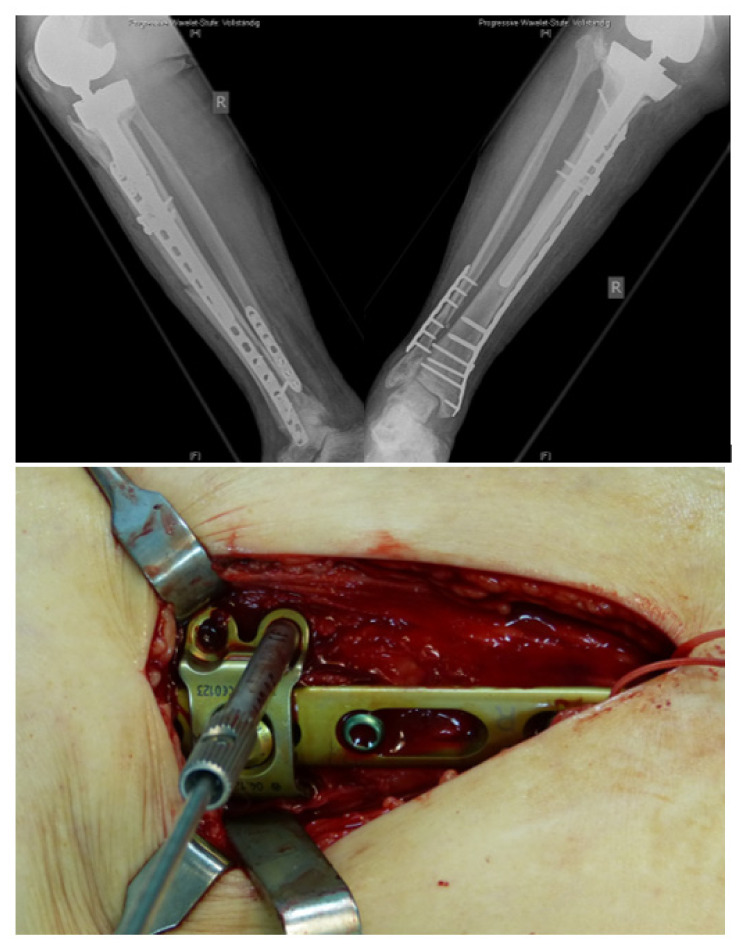
Locking attachment plate (Depuy-Synthes, Solothurn, Switzerland).

**Figure 2 jcm-11-01184-f002:**
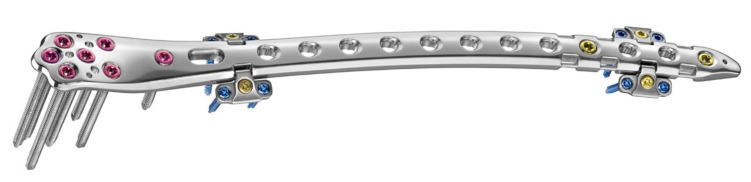
LOQTEQ^®^ VA Periprosthetic Plate (aap Implantate AG, Berlin, Germany).

**Figure 3 jcm-11-01184-f003:**
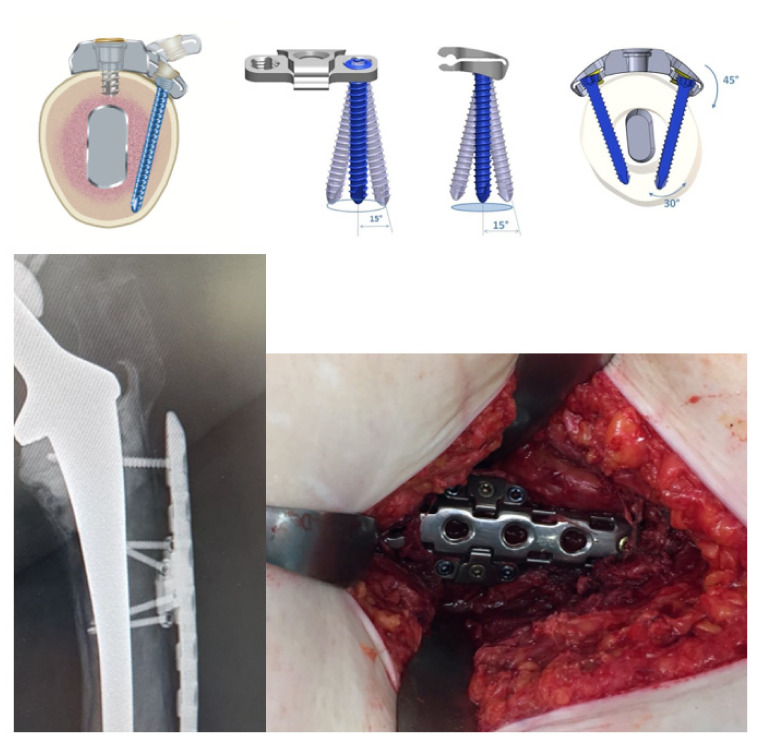
Hinge-technique with variable angle stable screw options, intraoperative and radiologic view.

**Figure 4 jcm-11-01184-f004:**
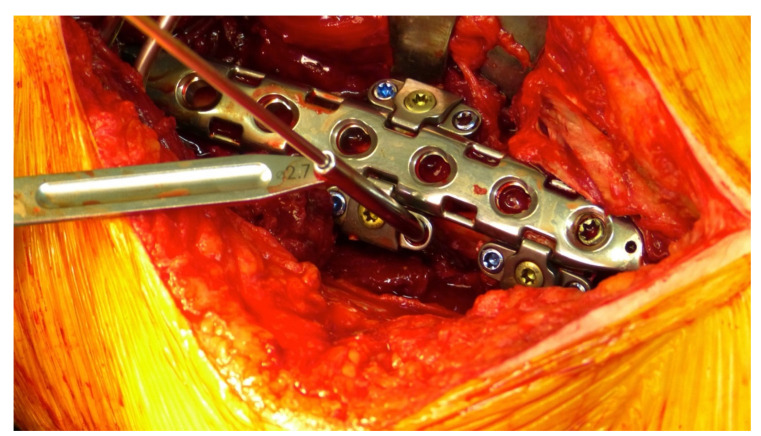
Freehand drill guide for 3.5 mm variable angle stable screws.

**Figure 5 jcm-11-01184-f005:**
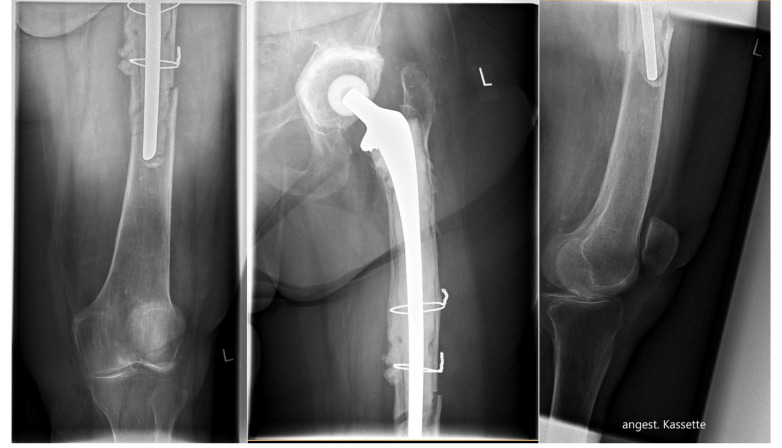
Periprosthetic fracture around a cemented total hip revision stem, postoperative x-rays of double plating using the hinge-technique for bicortical screw fixation, intraoperative view.

**Table 1 jcm-11-01184-t001:** Patient and study characteristics.

Age (Year)	80 (SD 11)	*n*
Sex	Female	22
	Male	4
ASA	ASA1	1
	ASA2	5
	ASA3	12
	ASA4	8
	ASA5	0
Fracture side	Left	14
	Right	12
UCS Classification	IV.3-B1	5x
	IV.3-C	3x
	V.3-B1	8x
	V.3-B3	1x
	V.3-C	3x
	V.3D	6x
Follow-up (month)	13.9 (SD 12)	

**Table 2 jcm-11-01184-t002:** Intra- and postoperative data.

Patient	OP-Time (min)	Double Plate	HHP *	KSS **	WOMAC ***	VAS ****	Complications	Fracture Healing
1	127	No	59	144	20	2		Yes
2	273	Yes	77	129	35	0		Yes
3	191	No	99	169	11	0		Yes
4	307	No	42	25	73	4	sec. dislocation	Yes
5	198	No	65	80	36	3		Yes
6	173	Yes	37	65	62	3		Yes
7	128	Yes	40	113	55	2		Yes
8	113	No	78	175	22	5	hematoma	Yes
9	125	No	94	194	17	2		Yes
10	113	Yes	90	169	13	0	hematoma	Yes
11	54	Yes	17	47	71	5		Yes
12	87	Yes	29	42	62	6		Yes
13	106	No	93	185	12	2		Yes
14	272	Yes	63	122	46	2		Yes
15	208	Yes	-	-	-	5	implant failure	No

* Harris Hip Score, ** Knee Society Score, *** Western Ontario and McMaster Universities Osteoarthritis Index, **** Visual Analogue Scale.

## Data Availability

Data supporting reported results can be provided by the authors.

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
