# Peer review of "LOQTEQ® VA Periprosthetic Plate—A New Concept for Bicortical Screw Fixation in Periprosthetic Fractures: A Technical Note"

_jcm, 2022, doi:10.3390/jcm11051184_

Round 1

Reviewer 1 Report

The manuscript " LOQTEQ® VA Periprosthetic Plate – a new concept for bicortical screw fixation in periprosthetic fractures. A technical note" introduces a new locking device for Periprosthetic fractures. The authors should be thanked for presenting their results with this novel device. 

Abstract: The abstract only mentions a new device is used. However, I think specifically introducing the plate directly in the abstract would be better. The background knowledge can be reduced, with more emphasis on the plate design and potential novelties of the device.

Introduction: The introduction section should further discuss the definition of a stable fracture, and the corresponding classifications commonly used. Also, the treatment options that are currently used in practice should also be elaborated. For instance, buttress plates, dynamic compression and LC plates etc all can be used and should be discussed regarding their shortcomings (biomechanics disadvantage and clinical outcome). 

patients and methods

general comments: how many surgeons performed the surgery and to what extent did they receive training (eg. fellowship trained etc). Tables should be constructed to better present preoperative and postoperative data. Eg. patient demographics should be presented in a table, and postoperative results should also be presented in a table. 

Line 83 "position of patient is similar to surgical technique of distal femoral plate" do you mean tibia? How is the incision made? Is it through minimally invasive and are all approaches standard?

Line 124-128 preoperative data and patient baseline characteristics are not presented. Comorbitides, ASA, and presence of infection etc. are all important data that should and need to be presented. 

Line 126 to 128 how do you define osteoporotic bone, shorts distal fracture fragment and loss of medial support?

Line 130-135 who recorded the patient reported outcomes? At what time points were the data recorded? Were the follow-ups and postoperative protocols standardize? 

Line 139-141: what statistical tests were used? 

Results:

Line 143-145 This should not be presented in your manuscript 

Line 147-158 the epidemiology of the patients are not described clearly. The initial implant designs should be included as well as the initial indication for arthroplasty surgery should also be described. A traumatic knee and OA knee is very different when it comes to postoperative fractures. How was an implant determined to be stable? Was it radiographic or any intraoperative findings that confirmed the stability of the implant?

Line 159-163 Why did the surgeons/authors decide to perform anterior plate as oppose to dual plating over medial and lateral aspect of the knee?

Line 165-168 Typing error in first line. How did you determine bone union? 

Line 172-177 Complications: were there any wound healing issues, considering the tendency for superficial infection and wound dehiscence over the proximal tibia, there should be some form of delayed wound healing or wound problem pertaining to the surgery. 

Discussion:

Line 186-189 main problems with current angular plate is placing screws across the implant. Are there intraoperative radiographic demonstrations showing that with the addition of hinge can precisely avoid the implants? In addition, are there radiographic images assuring that bicortical fixation can be achieved more frequently with the hinge?

Line 202-210 LOQTEQ has strong axial stiffness but does it pertain to the implant material or is it because of the hinge screws? From your abstract and introduction I assumed this was focused on the advantages of the hinge design, buy your discussion seems to focus on the overall implant. 

Line 229-231 Limitations such as no comparison group should also be mentioned. Since it is retrospective case series,  no significant conclusion can be drawn directly from this study.

Line 237-246 No significant conclusion should be made, therefore I would remove proven beneficial for the treatment of Periprosthetic fractures and reword this paragraph. 

Creating a table showing all 26 patient information should be supplemented. Table presenting patient baseline characteristics and results should also be presented. 

Reviewer 2 Report

This manuscript presents a topic of high clinical relevance. A newly developed plating system for periprosthetic fractures allows for bicortical screw fixation in variable angles. In this manuscript the surgical technique and first clinical results are presented.

In general this is a well written manuscript, but some aspects are not clear to the reader containing some inconsistencies. In general the discussion section needs a thorough revision: 

  • Line 19: In the Results section it is stated that you included 26 patients. Here you mention 25. Please be more precise and consistent. 
  • Line 20: Here you mention a follow up of 14.4 months, but in line 152 the mean follow up time was 13.9 months. Please explain and revise if appropriate. 
  • Line 109: Please add the respective torque value since there are two different according to the surgical guide (2.0 or 3.5 Nm).
  • Line 110: Do you have a reference for this biomechanical examinations? If yes, please cite it.
  • Line 139: Please name the statistical tests you used and what parameters you have compared. In the Results section the statistical analysis and statistical interpretation of your data is missing at all. Please explain and revise the Statistics of you manuscript.
  • Line 143-145: This is the general purpose of a Results section. I would suggest to delete these lines because this is redundant. 
  • Line 165: Either "This" or "The" examination.
  • Line 177: Please provide more details on the implant failure. Did the new Loqteq plate fail, or the prosthesis (cement augmentation)?
  • Line 179-185: This is a repetition of your introduction. The first paragraph of the Discussion section should be revised with the focus on your present study. I would suggest that lines 195-200 should be the first paragraph. 
  • Line 187: Please cite the Biomechanical studies you mention.
  • Line 211: How do you know, if the new Loqteq plate+anterior plate provides a higher stiffness? In the study by Wähnert et al. 2020 no additional anterior plate was tested. Please provide a reference that supports you assumption. Otherwise you cannot conclude that full weight-bearing is possible. (see line 242: you conclude that more research is needed to prove the superiority of the new plate).
  • Line 232: Why do you mention a different follow up time of 37 months compared to your study (13.9 or 14.4 months)? In my opinion 37 months is a rather long time for follow up, as the clinical follow ups are usually after 6, 12 or 24 months. Please explain your statement.
  • Line 248 and 249: Who is X.X.?
  • References: Please be consistent with formatting (either First Author et al., or name all authors)
  • References: Please check and, if appropriate, revise your Reference section for more recent literature published in the past 4-5 years. 17 out of 22 references are from 2015 or older.

Round 2

Reviewer 2 Report

All comments have been addressed and the manuscript has been revised respectively. Congratulations to this well written manuscript.
